# Random Blood Glucose, but Not HbA1c, Was Associated with Mortality in COVID-19 Patients with Type 2 Diabetes Mellitus—A Retrospective Study

Stefanus Gunawan Kandinata [1], Soebagijo Adi Soelistijo [2,*], Agung Pranoto [2] and Erwin Astha Triyono [3]

1   Department of Internal Medicine, Dr. Soetomo General Academic Hospital—Faculty of Medicine, Airlangga University, Surabaya 60132, Indonesia
2   Endocrinology, Metabolism and Diabetes Unit, Department of Internal Medicine, Dr. Soetomo General Academic Hospital—Faculty of Medicine, Airlangga University, Surabaya 60132, Indonesia
3   Tropical and Infectious Disease Unit, Department of Internal Medicine, Dr. Soetomo General Academic Hospital—Faculty of Medicine, Airlangga University, Surabaya 60132, Indonesia
*   Correspondence: soebagijo.adi.s@fk.unair.ac.id

**Abstract:** Previous studies have yielded inconsistent results on whether glycated hemoglobin (HbA1c) and random blood glucose (RBG) are associated with mortality of coronavirus disease 2019 (COVID-19) patients with type 2 diabetes mellitus (T2DM). This study aimed to assess the association of HbA1c and RBG with mortality among COVID-19 patients with T2DM. A retrospective study was conducted on 237 patients with COVID-19 and T2DM (survival ($n$ = 169) and non-survival groups ($n$ = 68)). Data on socio-demography, comorbidities, clinical symptoms, laboratory examination, and mortality were collected. Patients in the non-survival group had an older age range as compared with those in the survival group (60 (52.3–65.0) vs. 56.0 (48.5–61.5) years, $p$ = 0.009). There was no statistical gender difference between the two groups. After matching was done, chronic kidney disease, NLR, d-dimer, procalcitonin, and random blood glucose were higher in the non-survival group compared to the survival group ($p$ < 0.05). HbA1c levels were similar in survivors and non-survivors (8.7% vs. 8.9%, p=0.549). The level of RBG was independently associated with mortality of COVID-19 patients with T2DM ($p$ = 0.003, adjusted OR per 1-SD increment 2.55, 95% CI: 1.36–4.76). In conclusion, RBG was associated with the mortality of COVID-19 patients with T2DM, but HbA1c was not.

**Keywords:** diabetes; HbA1c; mortality; random blood glucose; COVID-19

## 1. Introduction

The coronavirus disease 2019 (COVID-19) pandemic has caused over 6.5 million deaths [1,2]. Vaccination programs have been carried out worldwide, but they still encounter challenges in implementation [3]. It is imperative that studies are conducted to improve in-hospital treatments for vulnerable groups, including individuals with type 2 diabetes mellitus (T2DM) who have a higher risk of severe COVID-19 [4–6]. A study revealed that even with the absence of other comorbidities, diabetic patients had a higher risk of severe pneumonia, higher hypercoagulable state, higher levels of tissue injury-related enzymes, and uncontrolled inflammatory response [7]. SARS-CoV-2 infection in T2DM could cause severe disease and higher mortality because of some close interactions: (1) hyperglycemia state in T2DM patients could modulate inflammatory responses and therefore predispose them to severe COVID-19; (2) SARS-CoV-2 infection in T2DM individuals could increase the reactive oxygen species (ROS) and inflammatory cytokines such as interleukin 6 and interferon-gamma, which could increase the chance for acute lung damage and acute respiratory distress syndrome (ARDS); (3) SARS-CoV-2 infection in T2DM individuals could induce insulin resistance and thereby induce hyperglycemia and vascular endothelial damages, leading to disseminated intravascular coagulation, thromboembolism, or

cardiovascular events; and (4) in diabetic patients, ACE-2 receptors, receptors for SARS-CoV-2 entry into host cells, are more prevalent, causing higher SARS-CoV-2 and therefore increased severity of pneumonia [8].

T2DM, one of the three types of diabetes, is concerning because its prevalence has been forecasted to increase from 6059 cases per 100,000 in 2017 to 7079 cases per 100,000 in 2030 [9]. T2DM has been associated with insulin resistance, a compensatory state due to a continuous increase in insulin production [9]. Elevated insulin will cause obesity, which in turn leads to insulin resistance—making a continuous cycle until pancreatic beta-cells inadequately meet the insulin demand [9]. This condition results in long-term hyperglycemia that leads to diabetic complications: macrovascular (cardiovascular disease, peripheral artery disease, or stroke), microvascular (nephropathy, neuropathy, or retinopathy), or miscellaneous complications, such as thyroid dysfunction [9].

In diagnosing T2DM, glycated hemoglobin (HbA1c) and random blood glucose (RBG) are robust indicators [10,11]. A correlation between HbA1c and mortality and morbidity in T2DM patients has been reported previously [12,13]. A systematic review conducted to assess whether HbA1c and RBG are associated with mortality of COVID-19 patients with T2DM [14] yielded inconsistent results. Therefore, this study aims to investigate the role of HbA1c and RBG as the mortality risk factors for COVID-19 patients with T2DM in the Indonesian population. Furthermore, a study evaluating the association between HbA1C or RBG and mortality has never been done in Indonesia to this date to the best of our knowledge.

## 2. Materials and Methods

### 2.1. Study Design and Participants

In this retrospective study, patients with COVID-19 and T2DM (*n* = 348), treated between October 2020 and March 2021 in Dr. Soetomo General Academic Hospital, Surabaya, Indonesia, were recruited. The inclusion criteria were: (1) adult patients (≥18 years old); (2) having T2DM as confirmed by International Statistical Classification of Diseases version 2010 (ICD-10) criteria; (3) having COVID-19 as confirmed by nasal swab reverse transcription polymerase chain reaction (RT-PCR); and (4) undergoing hospitalization in the isolation room. Patients diagnosed with leukemia, hemoglobinopathy, thalassemia, iron deficiency anemia, and hemolytic anemia were excluded from the study. The study protocol was approved by the Ethical Committee of Health Research of Dr. Soetomo General Academic Hospital (No. 0460/LOE/301.4.2/V/2021—Approval date: 5 May 2021) following the guidance from the Office for Human Research Protections (OHRP), under the requirement of the U.S. Department of Health and Human Services (HHS).

### 2.2. Study Measures

We collected and assessed the data on sociodemographics, comorbidities, clinical symptoms, laboratory examinations, and the outcome of the patients. Sociodemographic data included age and sex. The comorbidities recorded were previous diabetes mellitus history, cardiovascular disease, cerebrovascular disease, chronic kidney disease, and hypertension. The clinical symptoms observed were breathing difficulty, fever, cough, sore throat, aches and pain, diarrhea, anosmia, headache, and nausea. We also obtained laboratory examination results including neutrophil-to-lymphocyte ratio (NLR), platelet, RBG, HbA1c, D-dimer, fibrinogen, C-reactive protein (CRP), procalcitonin, and ferritin. Random blood glucose was measured as venous plasma glucose in mg/dL. The laboratory examinations were obtained from the initial hospital admission.

The severity of COVID-19 was assessed based on WHO 2021 criteria [15]. The patients were divided into two groups based on outcomes during hospitalization (survival vs. non-survival). Both of these groups were matched for age, gender, and severity of COVID-19.

### 2.3. Statistical Analysis

Descriptive statistics were presented in frequency (n) and percentage (%) for categorical variables and in mean ± SD for normally distributed continuous variables or median (interquartile range (IQR)) for skewed variables. Plausible associated factors with the outcome of COVID-19 patients with T2DM were determined using the $X^2$ test or Fisher exact test for categorical variables and the independent *t*-test or Mann–Whitney U test for continuous variables. A *p*-value of <0.05 was considered statistically significant. Significant results were followed by a logistic regression analysis to assess whether they were associated with mortality. The odds ratio (OR) and 95% confidence interval (CI) were calculated per 1-SD increment. All statistical analyses were performed on IBM SPSS Statistics 25.0 for Windows.

### 3. Results

#### 3.1. Characteristics of Patients with COVID-19 and T2DM

There were 348 patients admitted to Dr. Soetomo General Academic Hospital during the study period. Of them, 237 were included in the study. The clinical characteristics of the study population are presented in Table 1. A total of 169 (71.3%) patients survived and 68 (28.7%) died. There was a significant difference between the age of those who survived and those who died (56 (48.5–61.5) years old vs. 60 (52.3–65.0) years old, *p* = 0.009). No significant gender difference was found between the two groups (*p* = 0.140). Severe cases and breathing difficulty were more prevalent in the deceased group compared to the survivors (77.9% vs. 62.1%, *p* = 0.020; 82.4% vs. 62.8%, *p* = 0.026).

**Table 1.** Characteristics of COVID-19 patients with T2DM (unmatched).

| Variables | Category | Total (*n* = 237) | Survivors (*n* = 169, 71.3%) | Non-Survivors (*n* = 68, 28.7%) | *p*-Value |
|---|---|---|---|---|---|
| Age (years old), median (IQR) | | 56.0 (50.0–63.0) | 56.0 (48.5–61.5) | 60.0 (52.3–65.0) | 0.009 |
| Sex | Male | 125 (52.7%) | 84 (49.7%) | 41 (60.3%) | 0.140 |
| | Female | 112 (47.3%) | 85 (50.3%) | 27 (39.7%) | |
| Comorbidities | Cardiovascular diseases | 25 (10.5%) | 14 (8.3%) | 11 (16.2%) | 0.074 |
| | Cerebrovascular disease | 13 (5.5%) | 7 (4.1%) | 6 (8.8%) | 0.204 |
| | Chronic kidney disease | 61 (25.7%) | 31 (18.3%) | 30 (44.1%) | <0.001 |
| | Hypertension | 116 (48.6%) | 78 (46.2%) | 38 (55.9%) | 0.175 |
| Initial symptom | Breathing difficulty | 171 (72.2%) | 115 (68.0%) | 56 (82.4%) | 0.026 |
| | Fever | 120 (50.6%) | 83 (49.1%) | 37 (54.4%) | 0.460 |
| | Cough | 166 (70.0%) | 122 (72.2%) | 44 (64.7%) | 0.255 |
| | Sore throat | 11 (4.6%) | 8 (4.7%) | 3 (4.4%) | 1.000 |
| | Aches and pains | 9 (3.8%) | 6 (3.6%) | 3 (4.4%) | 0.719 |
| | Weak body | 36 (15.2%) | 23 (13.6%) | 13 (19.1%) | 0.285 |
| | Diarrhea | 22 (9.3%) | 18 (10.7%) | 4 (5.9%) | 0.253 |
| | Anosmia | 15 (6.3%) | 10 (5.9%) | 5 (7.4%) | 0.769 |
| | Headache | 1 (0.4%) | 1 (0.6%) | 0 (0%) | 1.000 |
| | Nausea | 36 (15.2%) | 29 (17.2%) | 7 (10.3%) | 0.183 |
| Disease severity | Non-severe | 79 (33.3%) | 64 (37.9%) | 15 (22.1%) | 0.020 |
| | Severe | 158 (66.7%) | 105 (62.1%) | 53 (77.9%) | |

After matching for age, gender, and case severity, the non-survival group had more chronic kidney disease than the survival group (30 (44.1%) vs. 18 (26.5%), *p* = 0.031). A significant difference between the two groups was not observed in other comorbidities (Table 2).

**Table 2.** Risk factors associated with mortality of COVID-19 patients with T2DM (matched).

| Variable | Category | Total (*n* = 136) | Survivors (*n* = 68) | Non-Survivors (*n* = 68) | *p*-Value |
|---|---|---|---|---|---|
| Comorbidity | Cardiovascular | 20 (14.7%) | 9 (13.2%) | 11 (16.2%) | 0.628 |
| | Cerebrovascular | 7 (5.1%) | 1 (1.5%) | 6 (8.8%) | 0.052 |
| | Chronic kidney disease | 48 (35.3%) | 18 (26.5%) | 30 (44.1%) | 0.031 |
| | Hypertension | 71 (52.2%) | 33 (48.5%) | 38 (55.9%) | 0.391 |
| Laboratory parameter | NLR | 7.2 (5.0–12.4) | 6.0 (4.1–9.6) | 9.1 (5.9–15.0) | 0.001 |
| | Platelet, $\times 10^3/\mu L$ | 229.5 (190.0–305.3) | 229.0 (191.0–299.0) | 228.0 (179.0–284.0) | 0.913 |
| | D-dimer, ng/mL | 1310.0 (770.0–3780.0) | 870.0 (500.0–2120.0) | 2225.0 (1080.0–14,270.0) | <0.001 |
| | Fibrinogen, mg/dL | 520.9 ± 180.1 | 543.5 ± 157.2 | 494.3 ± 202.2 | 0.038 |
| | CRP, mg/dL | 7.9 (3.7–13.8) | 6.9 (3.1–11.6) | 8.9 (5.8–14.1) | 0.165 |
| | Procalcitonin, ng/mL | 0.3 (0.1–0.5) | 0.2 (0.1–0.3) | 0.4 (0.2–1.0) | <0.001 |
| | Ferritin, ng/mL | 1014.8 (574.0–1716.4) | 886.9 (492.0–1511.7) | 1080.2 (629.2–2167.5) | 0.103 |
| | Random blood glucose, mg/dL | 218.5 (150.8–323.8) | 206.0 (126.0–289.5) | 243.5 (177.0–375.5) | 0.002 |
| | HbA1c, % | 8.8 (7.0–11.5) | 8.7 (7.4–11.6) | 8.9 (6.9–11.5) | 0.549 |

In regard to the laboratory parameters, the non-survival group had a significantly higher level of NLR (9.1 (5.9–15.0) vs. 6.0 (4.1–9.6), *p* = 0.001), d-dimer (2225 (1080–14,270) vs. 870 (500–2,120), *p* < 0.001), and procalcitonin (0.4 (0.2–1.0) vs. 0.2 (0.1–0.3), *p* < 0.001) (Table 2). In contrast, the survival group showed significantly higher fibrinogen levels than the deceased group (543.5 ± 157.2 vs. 494.3 ± 202.2, *p* = 0.038) (Table 2).

*3.2. Association of HbA1c Level with Mortality of COVID-19 Patients with T2DM*

The results of the analysis of HbA1c level as a mortality predictor in COVID-19 patients with T2DM is shown in Table 2. HbA1c value was similar in survivors and non-survivors (8.7% (7.4–11.6) vs. 8.9% (6.9–11.5), *p* = 0.549).

*3.3. Association of Random Blood Glucose with Mortality of COVID-19 Patients with T2DM*

There was a significant difference between survivors and non-survivors (206.0 (126.0–289.5) mg/dL vs. 243.5 (177.0–375.5) mg/dL, *p* = 0.002) (Table 2).

In the univariate logistic regression analysis, a higher RBG level was associated with a higher risk of mortality (OR 2.05, 95% CI 1.40–3.68, *p* = 0.002). After adjustment for potential confounders, the association between RBG and mortality remained significant (adjusted OR per 1-SD increment 2.55, 95% CI: 1.36–4.76, *p* = 0.003) (Table 3).

**Table 3.** Multivariate regression analysis of factors associated with mortality.

| Model | Per 1-SD Increment | |
|---|---|---|
| | OR (95% CI) | *p*-Value |
| Random blood glucose, crude | 2.05 (1.40–3.68) | 0.002 |
| Model 1 | 2.16 (1.30–3.58) | 0.003 |
| Model 2 | 2.08 (1.24–3.50) | 0.006 |
| Model 3 | 2.32 (1.33–4.03) | 0.003 |
| Model 4 | 2.37 (1.33–4.21) | 0.003 |
| Model 5 | 2.79 (1.51–5.17) | 0.001 |
| Model 6 | 2.55 (1.38–4.71) | 0.003 |

**Table 3.** *Cont.*

| Model | Per 1-SD Increment | |
| --- | --- | --- |
| | OR (95% CI) | *p*-Value |
| Model 7 | 2.55 (1.37–4.73) | 0.003 |
| Model 8 | 2.55 (1.36–4.76) | 0.003 |

Logistic regression analyses were performed to assess the association between RBG with in-hospital mortality of COVID-19 with T2DM. Multivariable-adjusted model 1 included adjustment for d-dimer; model 2 adjusted for variables in model 1 and procalcitonin; model 3 adjusted for variables in model 2 and NLR; model 4 adjusted for variables in model 3 and chronic kidney disease; model 5 adjusted for variables in model 4 and fibrinogen; model 6 adjusted for variables in model 5 and ferritin; model 7 adjusted for variables in model 6 and CRP; and Model 8 adjusted for variables in model 7 and cerebrovascular disease.

## 4. Discussion

This present study was conducted to determine the association of HbA1c and RBG with the mortality of COVID-19 patients with T2DM. During our initial analysis, we found that age was associated with the mortality of COVID-19 patients with T2DM. Previous studies have also revealed that age is one of the mortality-determining factors in COVID-19 and T2DM patients [13,16]. Although some studies found an association between sex and the mortality of COVID-19 cases with T2DM, the findings were not consistent [16,17]. A study found that among COVID-19 patients who had T2DM and cardiovascular disease, sex was not associated with in-hospital mortality [18]. In this present study, sex was also not associated with COVID-19 outcomes. In addition, among the comorbidities in the COVID-19 and T2DM patients (cardiovascular diseases, cerebrovascular diseases, chronic kidney diseases, and hypertension), we only found an association between chronic kidney disease and mortality. This finding is in agreement with two previous studies that found that COVID-19 patients with T2DM and kidney diseases had a significantly higher chance of death [16,19]. T2DM patients with diabetic nephropathy show chronic systemic inflammation that contributes to immunosuppression, which determines morbidity and mortality [20].

We found that breathing difficulty was significantly associated with mortality. A retrospective study in Wuhan, where a total of 153 patients were included, found a similar result [21]. Furthermore, some laboratory indicators such as NLR, d-dimer, and procalcitonin levels were associated with the mortality of patients. Several previous studies have also reported similar findings, including those conducted in China [21–23] and Iran [18], reflecting a more significant inflammatory response, hypercoagulable state, endothelial injury, and coinfection in the deceased group. However, we noted that the fibrinogen level was lower in the non-survivor group. This finding was similar to the CORONADO study, which observed slightly lower fibrinogen levels in the deceased group [24]. Furthermore, a study by Guo [7] found no statistically significant differences regarding fibrinogen levels in diabetic and non-diabetic patients infected with COVID-19. A considerable amount of fibrinogen data (26.5%) was found to be missing in the non-survivors, which is also thought to have influenced the results.

We found that HbA1c level was not statistically significant with mortality, suggesting it is not a predictor of mortality, which aligns with previous studies in France [24] and Austria [25]. One of the reasons for this is that there is no association between HbA1c level and COVID-19 severity, as reported by a previous study [26]. In addition, HbA1c as a mortality predictor is probably not significant because of its involvement in multiple factors, including roles in low-grade inflammation and immune cell functions, which are affected during SARS-CoV-2 infection [14]. In a previous study, the HbA1c level had statistical significance as a mortality prediction among COVID-19 and T2DM population only after the following classification was made: <7%, 7–8%, 8–9% and >9% [21]. We were unable to classify the HbA1c level based on this classification.

RBG level, however, was an independent risk factor for mortality in this present study. This finding is in agreement with that reported by a study carried out in Wuhan, China [21]. The level of capillary blood glucose on admission was found to be an independent mor-

tality risk factor among COVID-19 patients with T2DM [27]. There are some possible explanations of how RBG could be an independent risk factor for COVID-19 mortality in T2DM individuals: (1) high blood glucose in T2DM patients could modulate inflammatory responses leading to severe COVID-19 and predispose cytokine storm-associated death [8]; (2) increased glucose level could promote the replication of SARS-CoV-2 via mitochondrial ROS and hypoxia-inducible factor 1-alpha [28]; (3) high blood glucose might cause glucotoxicity that could lead to interstitial lung damage and therefore increase the risk of ARDS and death [8]; and (4) elevated blood glucose could also cause endothelial damage, which is associated with a risk of thromboembolic events such as pulmonary embolism [8]. Hence, blood glucose management during hospitalization is critical, as was suggested in a previous report [21]. Other than the in-hospital treatments, glycemic control could be achieved by improving patients' social support, self-efficacy, and self-care activities [29].

Despite the aforementioned literature in support of the findings of this present study, those who reported otherwise are worth noting. A study of 1004 diabetic patients with COVID-19 found that HbA1c or RBG was statistically associated with mortality [30]. Another study involving 3,295 COVID-19 patients found that HbA1c was associated with severe COVID-19, ICU admission, or all-cause mortality in both diabetic and non-diabetic groups [31]. A meta-analysis of nine trials concluded that a higher level of HbA1c was a parameter for higher mortality risk among COVID-19 and T2DM patients [32].

With the current number of samples used in this study, we have a limitation in classifying the ranges of HbA1c levels. Moreover, the HbA1c and RBG levels were only measured upon admission, whilst the progress of diabetes and the alteration of glycemic status during the hospitalization were not measured in this study.

## 5. Conclusions

HbA1c level had no association with mortality of COVID-19 patients with T2DM. In contrast, RBG level was an independent risk factor of mortality among individuals with SARS-CoV-2 infection and T2DM. Therefore, RBG level needs to be measured as early as possible in COVID-19-confirmed individuals with T2DM to establish early management to prevent mortality. For future studies, we suggest including a higher number of patients which could allow for a more specific classification of HbA1c level ranges.

**Author Contributions:** Conceptualization: S.G.K. and S.A.S.; methodology: S.G.K., S.A.S., A.P., and E.A.T.; software: S.G.K.; validation: S.A.S., A.P., and E.A.T.; formal analysis: S.G.K., S.A.S., A.P., and E.A.T.; investigation: S.A.S. and A.P.; resources: E.A.T.; data curation: S.G.K., S.A.S., A.P., and E.A.T.; writing—original data preparation: S.G.K. and S.A.S.; writing—review and editing: A.P. and E.A.T.; visualization: S.G.K.; supervision: S.A.S., A.P., and E.A.T.; project administration: S.G.K. and S.A.S. All authors have read and agreed to the published version of the manuscript.

**Funding:** This research received no external funding.

**Institutional Review Board Statement:** The study protocol was approved by the Ethical Committee of Health Research of Dr. Soetomo Hospital (No. 0460/LOE/301.4.2/V/2021—5 May 2021).

**Informed Consent Statement:** Written informed consent to publish this paper has been obtained from the patients.

**Data Availability Statement:** On behalf of future research, the raw data can be accessed by contacting the corresponding author.

**Conflicts of Interest:** The authors declare no conflict of interest.

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
