# Peer review of "Random Blood Glucose, but Not HbA1c, Was Associated with Mortality in COVID-19 Patients with Type 2 Diabetes Mellitus—A Retrospective Study"

_pathophysiology, doi:10.3390/pathophysiology30020012_

Round 1

Reviewer 1 Report

Major Points

1. This is quite a small study, which probably explains the lack of statistical significant associations for well-established risk factors such as presence of cardiovascular disease, hypertension and male sex.

2. Males are under-represented suggesting patient selection was biased.

3. The small number of males also makes it difficult to make any definitive conclusions on sex differences.

4. Rather than dichomotise most continuous variables, it would be more appropriate to compare means/medians of these variables between survivors and non-survivors and to perform logistic regression with mortality as the dependent variable and the variables of interest entered as (continuous) co-variates.

5. It is not clear why the authors chose a cut-off for serum ferritin of 2144.2.

6. In the Discussion, authors should mention a possible explanation of high blood glucose with mortality could be that high glucose is marker of a stress hormone response and therefore, of a more severe infection 

Minor Points

7. Can the authors confirm that they measured blood glucose rather than plasma glucose.

8. There is a typographical error in the sentence, 'Based on HbA1c level, patients were grouped into good glycemic 73 control (HbA1c level < 7% for <65 years old patients and HbA1c level < 8% for < 65 years 74 old patients)' (Section 2.2).co-variates.

9. The percentages of survivors and non survivors do not add up (Table 1).

Reviewer 2 Report

Title: Are HbA1c and random blood glucose associated with mortality of diabetic COVID-19 patients? – A retrospective study in Indonesia

Although study looks interesting there are issues with this manuscript.

1. Introduction:

-This section requires more details, mostly about type 2 diabetes, obesity, insulin resistance, and diabetes complications, and the authors should clarify the link between diabetes and COVID-19.

-The authors should clarify the novelty of this article in the ‘Introduction’ and ‘Conclusion’ section.

2. Materials and methods:

-The following sentence is unclear. It makes no difference and is instead repeated.

“Based on HbA1c level, patients were grouped into good glycemic control (HbA1c level < 7% for <65 years old patients and HbA1c level < 8% for < 65 years old patients)”

3. Results:

-Why there is so many variations in the results of various parameters of patients. Justify it. 

4. Discussion:

-How is this article more informative than the previously published ones? Justify it. 

5. Conclusion:

- This section is unclear to the reader. Authors should write in a concise manner, emphasizing on their outcomes and future prospects.                                                                                                                                                                                                                                                                                                                                                                                                                                                                                                                                                                                                                                                                                                                                                                                                                                                                                                                                                                                                                                                                                                                                                                                                                                                                                                                                                                                                                                                                                                                                                                                                                                                                                                                                                                                                                                                                                                                                                                                                                                                                                                                                                                                                                                                                                                                                                                                                                                                                                                                                                                                                                                                                                                                                                                                                                                                                                                                                                                                                                                                                                                                                                                                                                                                                                                                                                                                                                                                                                                                                                                                                                                                                                                                                                                                                                                                                                                                                                                                                         

Round 2

Reviewer 2 Report

The majority of the comments were implemented as suggested by the authors. However, some minor changes are still needed in the conclusion section.

Instead of beginning, "We did...........like that...", I would suggest beginning with novel findings.
